# Health-related quality of life and its predictors among chronic kidney disease patients: A hospital-based cross-sectional study

**Tasmia Tasnim[1], Kazi Muhammad Rezaul Karim📧[2]\*, Tanjina Rahman[2], Harun-Ur Rashid[3]**

**1** Department of Nutrition and Food Engineering, Daffodil International University, Daffodil Smart City, Birulia, Savar, Dhaka, Bangladesh, **2** Institute of Nutrition and Food Science, University of Dhaka, Dhaka, Bangladesh, **3** Kidney Foundation Hospital and Research Institute, Dhaka, Bangladesh

\* rkarim98@gmail.com, rezaul.infs@du.ac.bd

**Data Availability Statement:** All relevant data are within the manuscript and its Supporting information files.

## Abstract

Chronic kidney disease (CKD) has a significant impact on the health-related quality of life (HRQoL) of affected individuals due to its progressive and disabling nature. The aim of this study was to evaluate the HRQoL and its predictors among CKD patients. A cross-sectional study was carried out at kidney foundation Hospital and research Institute at Dhaka, Bangladesh. Kidney Disease Quality of Life (KDQoL™ -36) questionnaire were used to measure the HRQoL of CKD patients. The study also used kidney-targeted KDQoL-36 Summary Score (KSS). Socio-demographic and medical records were also collected. Descriptive statistics, and multiple linear regression were performed. Out of 430 patients, 77.9% were in moderate to advanced stage of CKD. Patients aged, occupation, income, co-morbidities such as diabetes or hypertension, medication used, and serum hemoglobin were found significantly ($p < 0.05$) associated at different stages of CKD. The mean domain scores of physical component summary (PCS), mental component summary (MCS), burden of kidney disease (BKD), effect of kidney disease (EKD), symptoms and problems of kidney disease (SPKD) subscales were 37.19, 45.94, 31.49, 63.95, and 73.35, respectively. The KSS was 63.24. The stage of CKD has been documented as an important predictor of HRQoL of all subscales of KDQoL-36 as well as KSS. The older age group also showed a clear link with a lower HRQoL in all subscales of KDQoL-36, except SPKD. In multiple linear regression analysis, stage of CKD, patients age, employment status and use of medication were found significant predictors of KSS. Further, higher levels of education, being married, absence of diabetes and heart disease were all independent predictors of a higher MCS. Whereas retirement, low duration of CKD and the use of three or more drugs strongly linked to worse scores of PCS. By addressing the determinants of poor HRQoL, healthcare providers can tailor treatment plans to better meet the needs of these individuals and ultimately enhance their overall well-being.

**Funding:** The project was completed under the research grant scheme for Dhaka University Teacher (2023-24) that is financially supported by the University Grant Commission (UGC), Bangladesh.

**Competing interests:** The authors have declared that no competing interests exist.

## Introduction

Chronic kidney disease (CKD) is associated with a high risk of morbidity and mortality, and presents a significant public health concern, with increasing incidence and prevalence [1]. Renal function is diminished at different levels in CKD, a multifaceted syndrome that begins at risk or injury and progresses through different stages of chronic kidney failure [2]. CKD was ranked as the 18th cause of global death, according to the Global Burden of Disease Study (GBDS) in 2010, which was 27th in 1990 [3]. With an estimated prevalence of >10% in 2022 worldwide [4], CKD is now understood to pose a serious risk to people's quality of life (QOL) as the condition advances. CKD has a significant impact on the health-related quality of life (HRQoL) of affected individuals due to its progressive and disabling nature [5]. Diabetes mellitus and cardiovascular disease are two other chronic non-communicable disorders that are diagnosed in a significant percentage of CKD cases [6], which can complicate a patient's physical and psychological needs and have further impact on HRQoL. Cardiovascular morbidity and early death are also linked to CKD, which also significantly lowers quality of life [7]. Quality of life is defined as "an individual's perception of their position in life in the context of the culture and value system where they live and in relation to their goals, expectations, standards, and concerns" [8]. However, HRQoL refers to QoL in the context of disease and health, which is a multidimensional concept that incorporates domains related to physical, mental, emotional, and social functioning. It is the subjective comparison between patients' expectations and reality for treatment and adaptation to the disease [9].

Individuals with CKD may experience limitations in their physical functioning, emotional well-being, and social interactions, leading to a decreased overall quality of life [10]. Low QoL has been the major problem in CKD patients, and its occurrence can adversely impact the course of the disease [11]. There are substantial differences in disease burden between the earlier and later stages of CKD, with patients experiencing significant reductions in HRQoL and life expectancy on progression to kidney failure [12]. From the early stages of the disease to its end stage, disabling symptoms, various food and social restrictions, fluid restrictions, and associated stigma and taboos affect the daily lives of the patients. Increased frequency and severity of symptoms as well as psychological distress are common in patients with CKD, with the advancement of CKD [13], and this ultimately affects HRQoL. Furthermore, a higher risk of cardiovascular diseases, including stroke, myocardial infarction, and heart failure prevails in patients with advanced CKD, leading to further healthcare resource utilization and economic burden [14], that pose significant challenges to healthcare systems, particularly in low-resource nations [15]. Upon reaching kidney failure, the burden of managing the disease through resource intensive renal replacement therapy (RRT) such as dialysis or transplant can further impact HRQoL and imposes both significantly high healthcare costs and major societal burden [16, 17].

Worldwide, the importance of including HRQoL indicators in the clinical management of CKD patients has been highlighted after several studies demonstrated the strong relationship between reduced HRQoL and increased morbidity and mortality [18, 19]. The collection of QoL data helps patients understand their disease and the implications of treatment on their health, thus determining their treatment goals and modalities [20]. The Kidney Disease Quality of Life (KDQoL) survey is a kidney disease-specific measure of HRQoL [21]. It includes both generic and disease-specific components to assess the physical health, mental health, and overall well-being of individuals with CKD. By utilizing the KDQoL survey for those with mild-to-moderate CKD, healthcare providers can better identify and address factors impacting HRQoL in this population. Bangladesh, a densely populated developing nation in Southeast Asia, is witnessing a notable increase in the annual prevalence of CKD. In contrast to the

world prevalence, a meta-analysis showed that the overall prevalence of CKD in Bangladesh alone is 22.48% [22], which is higher than the global rate. In 2018, kidney disease was the 9th leading cause of death in Bangladesh [23]. According to data from hospitals, urban areas, and disadvantaged communities, 11% of Bangladeshi people have CKD, with stage 3 or above [22, 24]. Previous studies have assessed HRQoL in people with CKD on dialysis [25, 26]. Despite advances in CKD care, HRQoL is often under assessed, particularly in resource-poor settings like Bangladesh, where financial constraints and limited healthcare access exacerbate patient outcomes. Also, no systematic evaluation of the HRQoL of the non-renal replacement therapy (RRT) CKD patients in Bangladesh has been carried out to date, thus ignoring the full burden of CKD. Therefore, this research gap highlights the need for further studies to assess HRQoL of CKD patients in Bangladesh.

Understanding the HRQoL in this group is crucial, as it can offer valuable insights into the challenges faced by individuals at various stages of CKD. By identifying the socio-demographic and clinical factors that influence HRQoL, healthcare providers can better tailor interventions to meet the unique needs of these patients, ultimately improving their well-being and potentially delaying CKD progression. Furthermore, insights from HRQoL assessments can be instrumental in shaping healthcare policies and interventions that more effectively address the needs of CKD patients at the community and primary care levels. Therefore, the aim of this study is to evaluate the underexplored area of HRQoL of non-RRT CKD patients in primary care settings in Bangladesh, a low-income country with distinct socio-economic and healthcare challenges. By using the KDQoL™-36 questionnaire, this study will comprehensively assess physical, mental, and kidney-specific HRQoL domains, providing insights from a resource-limited context where such data is sparse. Most existing studies have focused on high-income settings, making this research valuable for understanding HRQoL in a low-resource environment. Additionally, this study will investigate the correlations between HRQoL and socio-demographic and clinical factors across CKD stages, identifying predictors for targeted, stage-specific treatment strategies. By identifying these predictors, this study aims to provide a foundation for a bench-to-bedside framework to develop more personalized and cost-effective treatment approach that improve HRQoL, ensuring that CKD patients in Bangladesh receive both medical and psychosocial care. Ultimately, this research will guide future studies and help shape interventions that can reduce the burden of CKD and enhance the quality of life for patients in resource-limited settings.

## Methods

### Study design and subject

This cross-sectional study was conducted at the Kidney Foundation Hospital and Research Institute in Dhaka, Bangladesh, from September to December 2023. The study enrolled patients at all stages of chronic kidney disease (CKD) who were not undergoing renal replacement therapy (RRT). The inclusion criteria were adult CKD patients (≥18 years) with a confirmed diagnosis of CKD, as documented by a medical doctor, and who attended the hospital for follow-up care during the study period. Exclusion criteria included CKD patients who were receiving dialysis or had undergone kidney transplantation. Study participants were recruited from the Kidney Foundation Hospital and Research Institute, one of the leading referral centers for chronic kidney disease (CKD) care in Bangladesh. This specialized renal care facility, staffed by renowned nephrologists and consultants, serves a diverse patient population from both urban and rural areas, representing various socio-economic backgrounds and stages of CKD. The hospital accommodates approximately 250 to 300 outpatients daily, all of whom are Bangladeshi residents [27]. Given the hospital's broad demographic reach and its

role as a specialized treatment facility, our study participants can be considered representative of the national CKD population, encompassing a wide range of disease severities and enhancing the generalizability of the study's findings.

The sample size was calculated based on the formula N≥30k where k represents the number of predictors in the multiple regression model [28]. Considering 15 predictors in the model, a minimum sample size of 450 was required to ensure robust statistical analysis. A total of 450 patients were initially surveyed to meet this requirement. However, 20 questionnaires were excluded due to incomplete responses to the Kidney Disease Quality of Life (KDQoL) survey, resulting in a final sample size of 430 participants. A convenience sampling method was used to select participants. All eligible CKD patients who met the inclusion criteria and were not on RRT during the study period were approached for participation. Those who consented to participate were invited to complete the KDQoL questionnaire. The final sample size of 430 participants accurately reflects the patient population attending the Kidney Foundation Hospital during the study period and meets the required sample size for statistical analysis.

## Data collection tools

Face-to-face interviews were conducted with 430 CKD patients. Data were collected using structured questionnaire, which contained socio-demographic characteristics (age, sex, education, income, and other socio-economic factors), clinical parameters or medical record and Kidney Disease Quality of Life 36-item survey (KDQoL™-36). Interviews were conducted by trained university graduates, who were extensively prepared to ensure consistency, accuracy, and a neutral approach. Their training emphasized creating a non-judgmental environment and avoiding leading questions, ensuring participants felt comfortable and responded honestly.

Clinical data were gathered using standardized procedures. Information was extracted from the patients' medical records, including recent biochemical data such as serum creatinine levels (from the last 20 days prior to the interview). Clinical history such as diabetes mellitus, hypertension, cardiovascular disease, and the duration of chronic kidney disease was also acquired. BMI (kg/m2) was calculated by dividing participants' weight (in kilograms) by the square of their height (in meters). BMI was categorized into four groups which are underweight (≤18.5 kg/m2), normal weight (18.5–22.99 kg/m2), overweight (23.0–27.49 kg/m2) and obese (≥27.5 kg/m2) according to Asian BMI standards [29]. Anemia was defined as hemoglobin level of less than 12 g/dl as suggested by the KDIGO guidelines [30]. After explaining the purpose of the survey to the participants, those who were willing to take part in the study gave their written informed consent, before the interview.

## CKD stages

We estimated glomerular filtration rate (GFR) using the formulae of Chronic Kidney Disease Epidemiology Collaboration (CKD-EPI, 2021) [31]. CKD stages were classified according the level of estimated GFR (eGFR) expressed as mL/min/1.73 $m^2$: Stage 1 (eGFR of ≥90 mL/min/1.73 $m^2$), Stage 2 (eGFR of ≥ 60–89), Stage-3 (stage 3a - eGFR of ≥ 30–44, stage 3b- eGFR of ≥ 45–59); Stage-4 (eGFR of ≥15–29) and Stage 5 (eGFR of <15. Further the CKD patients were categorized into 3 groups initial/early (stage 1 & 2), moderate (stage 3a and 3b) and advanced stage (stage 4 &5).

## Quality of life: Kidney Disease Quality of Life

Kidney Disease Quality of Life 36-item survey (KDQoL™-36) was administered to measure health-related quality of life (HRQoL) of CKD patients [21]. The KDQoL-36 version has the

24-item kidney disease specific questionnaire as well as the 12-item Short-Form Health Survey (SF-12). The KDQoL-36 comprises five subscales calculated separately: (1) SF-12 physical component summary (PCS), (2) SF-12 mental component summary (MCS), (3) burden of kidney disease (BKD, 4 items), (4) symptoms of kidney disease (SPKD, 12 items), and (5) effects of kidney disease (EKD, 8 items) [32]. The SF-12 subscales (PCS and MCS) are scored using a condensed scale, with responses ranging from 2 to 6 points, and are combined into a single PCS and MCS score. For the kidney disease-targeted subscales (SPKD, BKD, and EKD), which assess the burden, symptoms, and effects of CKD, all items have 5 response options. The possible range of scores for each subscale is from 0 to 100, with lower scores indicating poor self-reported quality of life and higher scores indicate better HRQoL [33]. A higher PCS or MCS score represents better physical and mental health, while higher scores on the BKD, SPKD, and EKD subscales indicate less burden, fewer symptoms, and fewer effects of kidney disease. The study also used the KDQoL-36 Summary Score (KSS) for the kidney-targeted KDQoL-36 scales (Burdens of Kidney Disease, Symptoms and Problems of Kidney Disease, and Effects of Kidney Disease) [34]. KSS is the composite score, which may be useful when kidney-targeted health-related quality of life needs to be summarized in a single score [34]. The KDQOL-36TM Scoring Program (v2.0), an Excel scoring tool created by the Kidney Disease Quality of Life Working Group, was used for calculating scores of all subscales [32]. This program is available free for download online (http://www.rand.org/health/surveys_tools/kdqol.html).

## Ethical approval

The Declaration of Helsinki's recommendations were followed in conducting this study, and the study protocol was evaluated and approved by Ethical Review Committee of the Faculty of Biological Sciences, University of Dhaka, Dhaka, Bangladesh (Ref. No.239/Biol. Scs., Date: August 30, 2023). After explaining the purpose of the survey to the participants, those who were willing to take part in the study gave their written informed consent, before the interview.

## Statistical method

All analyses were performed with IBM SPSS version 21 (Armonk, NY: IBM Corp.) and Stata version 13.0 (Stata Corp, College Station, TX, USA). While continuous variables were shown as mean ± SD, categorical variables were given as frequency and percentage. Depending on analysis, several potential contributing factors (such as stage of CKD, age, employment status, education level and BMI) were divided into two or more groups. We have some dichotomized variables such as income (0 = Lower income <20000 TK, 1 = higher income Higher income ≥ 20000 Tk), marital status [1 = married, 0 = other (unmarried, widow, separate)], duration of CKD (0 = less duration <24 month, 1 = long duration ≥ 24 months), present of diabetics, hypertension and CVD (yes = 1, no = 0) and medication used (0 = ≥3 Medication, 1 = <3 Medication). An independent sample T-test and one-way ANOVA were performed to compare two groups and three or more groups in the analysis of CKD stages and KDQoL, respectively. Serum sodium, potassium and hemoglobin data were excluded from the multiple regression due to absence of data for all patients. Variables included in the regression model had a significant level of $p<0.10$ both in t-test or ANOVA. The model's multicollinearity was checked using the correlation of the independent variables and also the variance inflation factor (VIF) to remove the bias or confounder. Six multiple linear regression was employed to identify the potential predictors of the physical and mental component summary, as well as kidney specific domain such as Effects of Kidney Disease, Symptoms and Problems of Kidney

Disease, Burdens of Kidney Disease and KDQoL-36 Summary Score. P value < 0.05 considered as statistically significant.

## Results

### Socio-demographic characteristics of the CKD patients

Out of 430 participants 55.1% were male. About half (49.5%) of the study participants were in the age group 45–64 years. Three-fourths (76.3%) of CKD patients were married, 87.5% were literate, and only 17.7% were in job service (Table 1). Around two-thirds (60.2%) of the study subject have the monthly income below 20000 TK. Total CKD patients were classified into five stages such as: 10.5% in stage 1, 11.6% in stage 2, 11.4% in stage 3a, 28.4% in stage 3b, 27.2% in stage 4 and 10.9% in stage 5. Later the study subjects were categorized into 3 groups: early/ initial stage 22.1% (CKD stage 1&2), moderate stage 39.8% (CKD stage 3) and advance stage 38.1% (CKD stage 4&5). More than half (59.2%) of the study participants suffered from CKD for less than 2 years. The most common comorbidity of CKD was hypertension, accounting for 73.7%, followed by diabetes mellitus in 36.8% and CVD in 15.8% (Table 1). In term of nutritional status, 8% CKD patients were underweighted, 37.9% were in normal weight, 38.2% were in overweight and 15.9% were in obese according to Asian cutoff point of BMI (Table 1).

**Table 1. Socio-demographic and clinical characteristics of participants of chronic kidney disease patients.**

| Characteristic | N (%) | Characteristic | N (%) |
|---|---|---|---|
| **Gender** | | **CKD Stage** | |
| Male | 237 (55.1) | Stage 1 | 45 (10.5) |
| Female | 193 (44.9) | Stage 2 | 50 (11.6) |
| **Age (Years)** | | Stage 3a | 49 (11.4) |
| 18–44 | 147 (34.2) | Stage 3b | 122 (28.4) |
| 45–64 | 213 (49.5) | Stage 4 | 117 (27.2) |
| ≥65 | 70 (16.3) | Stage 5 | 47 (10.9) |
| **Marital Status** | | **Duration of CKD** | |
| Married | 328 (76.3) | Low (<24 months) | 254 (59.1) |
| Others (unmarried, separated, Widowed) | 102 (23.7) | Long (≥24 months | 176 (40.9) |
| **Occupation** | | **Co-morbidity** | |
| Job Service | 76 (17.7) | Diabetics | 166 (38.6) |
| Housewife | 136 (31.6) | Hypertension | 317 (73.7) |
| Farmer/Day Laborer | 87 (20.2) | CVD | 68 (15.8) |
| Retired | 21 (4.9) | **Medication used** | |
| Others (Student, unemployed) | 110 (25.6) | <3 Medication | 265 (61.6) |
| **Educational status** | | ≥3 Medication | 165 (38.4) |
| Illiterate | 54 (12.6) | **Nutritional Status (BMI, Asian cut off value)** | |
| Primary | 175 (40.7) | Underweight (BMI <18.5) | 34 (8) |
| Secondary | 113 (26.3) | Normal (BMI 18.5–22.99) | 162 (37.9) |
| Higher Education | 88 (20.5 | Overweight (BMI 23–27.49) | 163 (38.2) |
| **Income** | | Obese (BMI ≥27.5) | 68 (15.9) |
| Lower income (<20000 TK) | 259 (60.2) | Serum Hb (n = 226, mean ± SD) | 10.67 ±1.95 |
| Higher income (≥ 20000 Tk) | 171 (39.8) | Serum Na (mmol/L) (n = 367, mean ± SD) | 138.45 ± 7.8 |
| Family History of CKD (yes) | 81 (18.8) | Serum K (mmol/L) (n = 366, mean ± SD) | 4.34 ± 0.68 |

CKD: chronic kidney disease, CVD: cardiovascular disease, BMI: body mass index, Hb: hemoglobin, Na: Sodium, K: potassium

## Distribution of socio-demographic and clinical characteristics at different stages of CKD

Table 2 describes the distribution of CKD patients according to different stages of CKD, and association with various sociodemographic variables. CKD stages were found significantly associated in different aged groups. Aged groups 45–64, and ≥65year were more vulnerable to moderate to advanced stages of CKD. Patients' occupation, income, presence or absence of diabetes or hypertension, medication used, and serum hemoglobin were found significantly associated at different stages of CKD by $\chi^2$ square test (Table 2). Whereas serum potassium also found significant difference in different stages of CKD by one-way ANOVA.

## Association between Kidney Disease Quality of Life (KDQoL™ -36) and patient characteristics

Table 3 illustrates the findings of the statistical comparison of the KDQoL-36 domains of patients with chronic kidney disease (CKD) based on categorical socio-demographic characteristics. The KDQoL-36 comprises five subscales, physical component summary (PCS), mental component summary (MCS), burden of kidney disease (BKD), symptom/problem of kidney disease (SPKD), and effects of kidney disease (EKD). The mean scores for the BKD, SPKD, EKD, PSC, and MSC were 31.49 ± 13.6, 73.35 ± 13.2, 69.95 ± 11.2, 37.19 ± 8.1 and 45.94 ± 7.5 respectively (Table 3). The mean composite score of kidney-targeted health-related quality of life (KSS) was 63.24 ± 10.0 (Table 3). Quality of life, as measured by the KDQoL-36, mean scores in each subscale gradually decreased from early/initial stage of CKD to advanced stage of CKD (Fig 1). The scores were significantly (p<0.05) lowest in the advanced stage (stage 4 & 5) for all subscales. That indicates that there is an inverse association between health-related quality of life (HRQoL) and stages of CKD. Female groups were found lower score in all subclasses of HRQoL to their counterpart. Quality of life significantly decreased in higher aged groups (45–64, and ≥65 aged groups) (Table 3). Higher education status and employment status were positively associated with quality of life. Absence of comorbidity such as diabetics, hypertension and CVD were also positively associated with KDQoL scores (Table 3). All KDQoL dimensions showed significantly higher scores, where patients had normal serum hemoglobin concentrations (serum hemoglobin ≥12 g/dl) but the exception of burden of renal disease (Table 3).

## Multiple linear regression analysis for predictors of KDQoL-36 scales

Multiple linear regression models were applied to see which factors influence the different components of the KDQoL-36 (Tables 4 and 5). Only those variables in table-3 had a significant level of $p<0.10$ (both in t-test or ANOVA) were included in the multiple linear regression analysis. Serum hemoglobin was excluded from multiple regression model due to insufficient numbers of data.

After adjustment through multiple linear regression, stage of CKD, patients age, employment status and duration of CKD were found significant predictors of Burden of Kidney Disease (Table 4). Advanced stage of CKD (stage 4 & 5), aged groups ≥ 65 years, long duration of CKD, and housewife/unemployed/students' groups were significantly negatively associated with the respective reference group (Table 4). Like BKD, advanced stage of CKD, occupation level (retired/housewife/others), education status (primary level), long duration of CKD, present of CVD, and ≥3 medication, were found significant potential predictors of symptom/problem of kidney disease (Table 4). Five variables such as stages of CKD, aged groups, marital status, duration of CKD, and number of medications used were found significant potential

**Table 2. Distribution of socio-demographic and clinical characteristics at different stages of chronic kidney disease.**

| Variable | Stage of CKD | | | P value |
|---|---|---|---|---|
| | Stage 1&2 | Stage 3 | Stage 4&5 | |
| **Gender** | | | | |
| Male | 54 (22.8) | 102 (43.0) | 81 (34.2) | 0.157 |
| Female | 41 (21.2) | 69 (35.8) | 83 (43.0) | |
| **Age (Years)** | | | | |
| 18–44 | 51 (34.7) | 53 (36.1) | 43 (29.3) | <0.001 |
| 45–64 | 36 (16.9) | 85 (39.9) | 92 (43.2) | |
| ≥65 | 8 (11.4) | 33 (47.1) | 29 (41.4) | |
| **Marital Status** | | | | |
| Married | 73 (22.3) | 130 (39.6) | 125 (38.1) | 0.988 |
| Others (unmarried, separated, Widowed) | 22 (21.6) | 41 (40.2) | 39 (38.2) | |
| **Occupation** | | | | |
| Job Service | 24 (31.6) | 33 (43.4) | 19 (25.0) | 0.011 |
| Housewife | 29 (21.3) | 40 (29.4) | 67 (49.3) | |
| Farmer/Day Laborer | 18 (20.7) | 35 (40.2) | 34 (39.1) | |
| Retired | 3 (14.3) | 12 (57.1) | 6 (28.6) | |
| Others (Student, unemployed) | 21 (19.1) | 51 (46.4) | 38 (34.5) | |
| **Educational status** | | | | |
| Illiterate | 6 (11.1) | 24 (44.4) | 24 (44.4) | 0.333 |
| Primary | 38 (21.7) | 66 (37.7) | 71 (40.6) | |
| Secondary | 26 (23.0) | 47 (41.6) | 40 (35.4) | |
| Higher Education | 25 (28.4) | 34 (38.6) | 29 (33.0) | |
| **Income** | | | | |
| Lower income (<20000 TK) | 63 (24.3) | 90 (34.7) | 106 (40.9) | 0.031 |
| Higher income (≥ 20000 Tk) | 32 (18.7) | 81 (47.4) | 58 (33.9) | |
| **BMI (Asian cut off value)** | | | | |
| Underweight (BMI <18.5) | 7 (20.6) | 15 (44.1) | 12 (35.3) | 0.523 |
| Normal (BMI 18.5–22.99) | 37 (22.8) | 59 (36.4) | 66 (40.7) | |
| Overweight (BMI 23–27.49) | 29 (17.8) | 71 (43.6) | 63 (38.7) | |
| Obese (BMI ≥27.5) | 20 (29.4) | 25 (36.8) | 23 (33.8) | |
| **Duration of CKD** | | | | |
| Low (<24 months) | 56 (22.0) | 112 (44.1) | 86 (33.9) | 0.052 |
| Long (≥24 months | 39 (22.2) | 59 (33.5) | 78 (44.3) | |
| **Family History of CKD** | | | | |
| Yes | 14 (17.3) | 34 (42.0) | 33 (40.7) | 0.511 |
| No | 81 (23.2) | 137 (39.3) | 131 (37.5) | |
| **Diabetics** | | | | |
| Yes | 27 (16.3) | 64 (38.6) | 75 (45.2) | 0.021 |
| No | 68 (25.8) | 107 (40.5) | 89 (33.7) | |
| **Hypertension** | | | | |
| Yes | 48 (15.1) | 129 (40.7) | 140 (44.2) | <0.001 |
| No | 47 (41.6) | 42 (37.2) | 24 (21.2) | |
| **CVD** | | | | |
| Yes | 8 (11.8) | 30 (44.1) | 30 (44.1) | 0.080 |
| No | 87 (24.0) | 141 (39.0) | 134 (37.0) | |
| **Medication used** | | | | |

(*Continued*)

**Table 2.** (Continued)

| Variable | Stage of CKD | | | P value |
|---|---|---|---|---|
| | Stage 1&2 | Stage 3 | Stage 4&5 | |
| <3 Medication | 73 (27.5) | 101 (38.1) | 91 (34.3) | 0.002 |
| ≥3 Medication | 22 (13.3) | 70 (42.4) | 73 (44.2) | |
| **Serum Hb (n = 226)** | | | | |
| <12 g/dl | 18 (10.7) | 66 (39.3) | 84 (50.0) | <0.001 |
| ≥ 12 g/dl | 21 (36.2) | 28 (48.3) | 9 (15.5) | |
| Serum Na (mmol/L) (n = 367, mean ± SD) | 139.26 ± 4.1 | 139.13 ± 4.2 | 137.34 ± 11.3 | 0.092 |
| Serum K (mmol/L) (n = 366, mean ± SD) | 4.20 ± 0.50 | 4.28 ± 0.67 | 4.47 ± 0.73 | 0.008 |

CKD: chronic kidney disease, CVD: cardiovascular disease, BMI: body mass index, Hb: hemoglobin, Na: Sodium, K: potassium, SD: standard deviation. Data is presented as n (%). *Chi square (χ2)* test was performed and *p <0.05* is considered significant.

predictors of Effect of kidney disease (Table 4). Four variable such as stages of CKD, advance aged group, occupation level (Housewife/student), and ≥3 medication, were found significant potential predictors of KSS (KDQoL-36 Summary Score) (Table 4).

The present study revealed that advanced stages of CKD (stage 4 &5), aged 65 and older, retired person, ≥3 medication used, and low duration of CKD (<24 months) were predictors of worse QoL in the physical component summary, after adjustments through multiple linear regression (Table 5). The early stages of CKD (stages 1 and 2), being married, being younger (18–44 years old), having a higher education level, and not having diabetes or cardiovascular disease complications were the independent predictors of a higher mental component summary (Table 5).

## Discussion

The current investigation presents data on the evaluation of the health-related quality of life among patients in Bangladesh with CKD stage 1–5 not on RRT using the KDQoL-36 questionnaire. A negative correlation between HRQoL and stages of CKD has been documented, and the severity of disease has been recognized as an important predictor of quality of life. The older age group also showed a clear and important link with a lower quality of life, impacting the burden and effect of kidney disease, as well as KSS, including physical and mental components. Higher levels of education above secondary school, a lack of diabetes, and the absence of heart disease were all correlated with an enhanced mental composite summary score (MCS). Retirement and the use of more drugs are also strongly linked to worse scores of physical component summary (PCS). This study highlights the urgent need for a bench-to-bedside framework to bridge the gap between research and clinical care, emphasizing the integration of routine HRQoL assessments and tailored interventions [35]. By addressing context- specific challenges, such as limited education and high rates of comorbidities, these strategies hold the potential to enhance patient outcomes and provide a scalable model for other low-resource settings.

According to this study, symptoms/problems of kidney disease (SPKD) and burden of kidney disease (BKD) were the highest and lowest HRQoL measures, respectively. Our results are in line with Alam *et al* [26] who indicated the lowest and highest mean scores of HRQoL to be BDK (16.93±13.0) and SPKD (81.09±13.14), respectively. On the other hand, Berhe *et al* [36] and Kim *et al* [37] showed SPKD to be the highest HRQoL measures among CKD patients in Ethiopia while PCS and MCS to be the lowest respectively. Senanayake *et al* [38] and

**Table 3. Quality of life of the participants according to KDQoL-36 domine with different variables.**

| Variable | BKD | SPKD | EKD | PSC | MCS | KSS |
|---|---|---|---|---|---|---|
| **Gender** | | | | | | |
| **Total score** | 31.49 ± 13.6 | 73.35 ± 13.2 | 63.95 ± 11.2 | 37.19 ± 8.1 | 45.94 ± 7.5 | 63.24 ± 10.0 |
| Male | 32.45 ± 14.0 | 74.05 ± 13.4 | 65.52 ± 11.6 | 37.39 ± 8.0 | 46.13 ± 7.4 | 64.27 ± 10.0 |
| Female | 30.31 ± 12.1 | 72.49 ± 12.8 | 62.03 ±10.4 | 36.94 ± 8.2 | 45.7 ± 7.6 | 61.97 ± 9.8 |
| P -value | 0.105 | 0.224 | 0.001 | 0.570 | 0.563 | 0.018 |
| **Age (Years)** | | | | | | |
| 18–44 | 35.95 ± 16.0 | 76.56 ± 11.4 | 68.6 ± 11.5 | 38.46 ± 8.4 | 47.82 ± 7.4 | 67.14 ± 10 |
| 45–64 | 30.48 ± 11.7 | 73.1 ± 13.0 | 62.88 ± 9.9 | 37.75 ± 7.4 | 45.85 ± 6.4 | 62.59 ± 8.9 |
| ≥65 | 25.17 ± 10.0 | 67.37 ± 15.1 | 57.45 ± 10.4 | 32.83 ± 7.9 | 42.26 ± 9.4 | 57.03 ± 9.5 |
| ANOVA, p-value | <0.001 | <0.001 | <0.001 | <0.001 | <0.001 | <0.001 |
| **Marital Status** | | | | | | |
| Married | 31.65 ± 13.1 | 73.88 ± 12.4 | 63.38 ± 10.8 | 37.46 ± 8.0 | 46.51 ± 6.9 | 63.34 ± 9.3 |
| Others (unmarried, separated, Widowed) | 30.98 ± 15.1 | 71.63 ± 15.4 | 65.78 ± 12.2 | 36.33 ± 8.1 | 44.08 ± 8.9 | 62.9 ± 11.9 |
| p-value | 0.665 | 0.133 | 0.061 | 0.220 | 0.004 | 0.700 |
| **Occupation** | | | | | | |
| Job Service | 35.27 ± 13.7 | 78.52 ± 9.2 | 66.57 ± 11.8 | 38.28 ± 8.1 | 47.79 ± 6.2 | 67.33 ± 8.1 |
| Housewife | 29.27 ± 12.2 | 73.02 ± 11.6 | 61.23 ± 10.3 | 37.12 ± 8.3 | 45.69 ± 7.4 | 61.8 ± 9.2 |
| Farmer/Day Laborer | 32.9 ± 12.7 | 75.41 ± 12.0 | 64.26 ± 11.3 | 37.0 ± 7.1 | 45.79 ± 7.4 | 64.61 ± 8.9 |
| Retired | 31.25 ± 11.1 | 66.55 ± 16.8 | 63.69 ± 9.2 | 31.7 ± 6.4 | 45.51 ± 6.6 | 59.71 ± 10.5 |
| Others (Student, unemployed) | 30.54 ± 15.5 | 69.83 ± 15.8 | 65.31 ± 11.7 | 37.7 ± 8.4 | 45.16 ± 8.4 | 61.78 ± 11.8 |
| ANOVA, p-value | 0.026 | <0.001 | 0.008 | 0.027 | 0.196 | <0.001 |
| **Educational status** | | | | | | |
| Illiterate | 28.0 ± 11.5 | 67.59 ± 14.5 | 59.72 ± 10.4 | 36.25 ± 8.4 | 42.71 ± 8.0 | 58.37 ± 10.5 |
| Primary | 30.52 ± 13.3 | 73.61 ± 12.5 | 63.25 ± 10.7 | 37.47 ± 7.8 | 45.8 ± 7.4 | 62.97 ± 9.4 |
| Secondary | 34.01 ± 14.6 | 73.83 ± 14.0 | 65.73 ± 10.7 | 36.91 ± 7.8 | 46.41 ± 7.4 | 64.49 ± 10.1 |
| Higher Education | 32.31 ± 13.6 | 75.74 ± 11.8 | 65.66 ± 12.7 | 37.58 ± 8.7 | 47.57 ± 7.0 | 65.14 ± 9.8 |
| ANOVA, p-value | 0.034 | 0.004 | 0.004 | 0.736 | 0.002 | <0.001 |
| **Income** | | | | | | |
| Lower income (<20000 TK) | 31.04 ± 13.4 | 74.43 ± 12.2 | 63.13 ± 11.5 | 37.85 ± 8.2 | 45.47 ± 7.7 | 63.43 ± 9.9 |
| Higher income (≥ 20000 Tk) | 32.16 ± 13.8 | 71.70 ± 14.4 | 65.20 ± 10.7 | 36.19 ± 7.7 | 46.64 ± 7.2 | 62.94 ± 10.1 |
| p-value | 0.406 | 0.036 | 0.061 | 0.037 | 0.114 | 0.622 |
| **BMI (Asian cut off value)** | | | | | | |
| Underweight (BMI <18.5) | 32.46 ± 17.7 | 75.20 ± 12.0 | 65.35 ± 12.5 | 37.28 ± 8.4 | 45.99 ± 7.9 | 64.79 ± 10.9 |
| Normal (BMI 18.5–22.99) | 32.87 ± 13.3 | 72.2 ± 14.0 | 63.75 ± 11.4 | 36.98 ± 7.7 | 45.52 ± 7.7 | 62.84 ± 10.5 |
| Overweight (BMI 23–27.49) | 30.86 ± 13.4 | 72.54 ± 13.8 | 63.76 ± 10.6 | 36.76 ± 8.5 | 46.24 ± 7.3 | 62.67 ± 10.0 |
| Obese (BMI ≥27.5) | 29.50 ± 12.3 | 76.73 ± 9.4 | 64.25 ± 11.5 | 38.78 ± 7.7 | 45.94 ± 7.4 | 64.71 ± 8.2 |
| ANOVA, p-value | 0.308 | 0.075 | 0.878 | 0.367 | 0.861 | 0.384 |
| **Duration of CKD** | | | | | | |
| Low (<24 months) | 34.62 ± 15.0 | 70.94 ± 14.0 | 65.66 ± 11.4 | 36.37 ± 8.3 | 46.4 ± 7.6 | 63.13 ± 10.8 |
| Long (≥24 months | 26.97 ± 9.6 | 76.82 ± 10.9 | 61.49 ± 10.5 | 38.37 ± 7.5 | 45.26 ± 7.3 | 63.4 ± 8.7 |
| p-value | <0.001 | <0.001 | <0.001 | 0.012 | 0.123 | 0.783 |
| **Diabetics** | | | | | | |
| Yes | 29.37 ± 12.1 | 68.59 ± 14.2 | 60.82 ± 10.3 | 34.98 ± 7.3 | 44.0 ± 7.7 | 59.46 ± 9.8 |
| No | 32.82 ± 14.3 | 76.34 ± 11.6 | 65.92 ± 11.2 | 38.58 ± 8.2 | 47.15 ± 7.11 | 65.61 ± 9.3 |
| p-value | 0.010 | <0.001 | <0.001 | <0.001 | <0.001 | <0.001 |
| **Hypertension** | | | | | | |
| Yes | 30.38 ± 13.2 | 71.09 ± 13.3 | 62.67 ± 11.0 | 36.13 ± 7.7 | 44.96 ± 7.5 | 61.5 ± 9.9 |

(*Continued*)

**Table 3.** (Continued)

| Variable | BKD | SPKD | EKD | PSC | MCS | KSS |
|---|---|---|---|---|---|---|
| No | 34.6 ± 14.1 | 79.68 ± 10.5 | 67.56 ± 11.0 | 40.17 ± 8.4 | 48.68 ± 6.7 | 68.13 ± 8.5 |
| p-value | 0.005 | <0.001 | <0.001 | <0.001 | <0.001 | <0.001 |
| **CVD** | | | | | | |
| Yes | 30.69 ± 15.3 | 63.2 ± 15.6 | 60.25 ± 11.5 | 33.43 ± 7.9 | 42.27 ± 8.5 | 56.8 ± 11.3 |
| No | 31.64 ± 13.2 | 75.25 ± 11.7 | 64.65 ± 11.0 | 37.9 ± 7.9 | 46.62 ± 7.1 | 64.42 ± 9.2 |
| p-value | 0.601 | <0.001 | 0.003 | <0.001 | <0.001 | <0.001 |
| **Medication used** | | | | | | |
| <3 Medication | 32.23 ± 13.4 | 79.16 ± 9.0 | 66.52 ± 10.4 | 39.42 ± 7.7 | 47.21 ± 6.8 | 67.1 ± 7.7 |
| ≥3 Medication | 30.3 ± 13.7 | 64.09 ± 13.6 | 59.83 ± 11.3 | 33.62 ± 7.3 | 43.89 ± 8.0 | 57.04 ± 10.1 |
| p-value | 0.153 | <0.001 | <0.001 | <0.001 | <0.001 | <0.001 |
| **Serum Hb (n = 226)** | | | | | | |
| <12 g/dl | 30.17 ± 13.5 | 68.8 ± 14.6 | 63.06 ± 10.2 | 35.39 ± 8.3 | 45.08 ± 7.7 | 60.45 ± 10.2 |
| ≥ 12 g/dl | 34.15 ± 16.25 | 78.68 ± 11.7 | 67.94 ± 12.3 | 39.61 ± 7.9 | 49.1 ± 7.7 | 67.68 ± 9.4 |
| p-value | 0.069 | <0.001 | 0.003 | 0.001 | 0.001 | <0.001 |

BKD: burden of kidney disease, SPKD: symptom/problem of kidney disease, EKD: effects of kidney disease, PCS: physical component summary, MCS: mental component summary, KSS: KDQoL-36 Summary Score, BMI: body mass index, CKD: chronic kidney disease, CVD: cardiovascular disease, Hb: hemoglobin, SD: standard deviation.

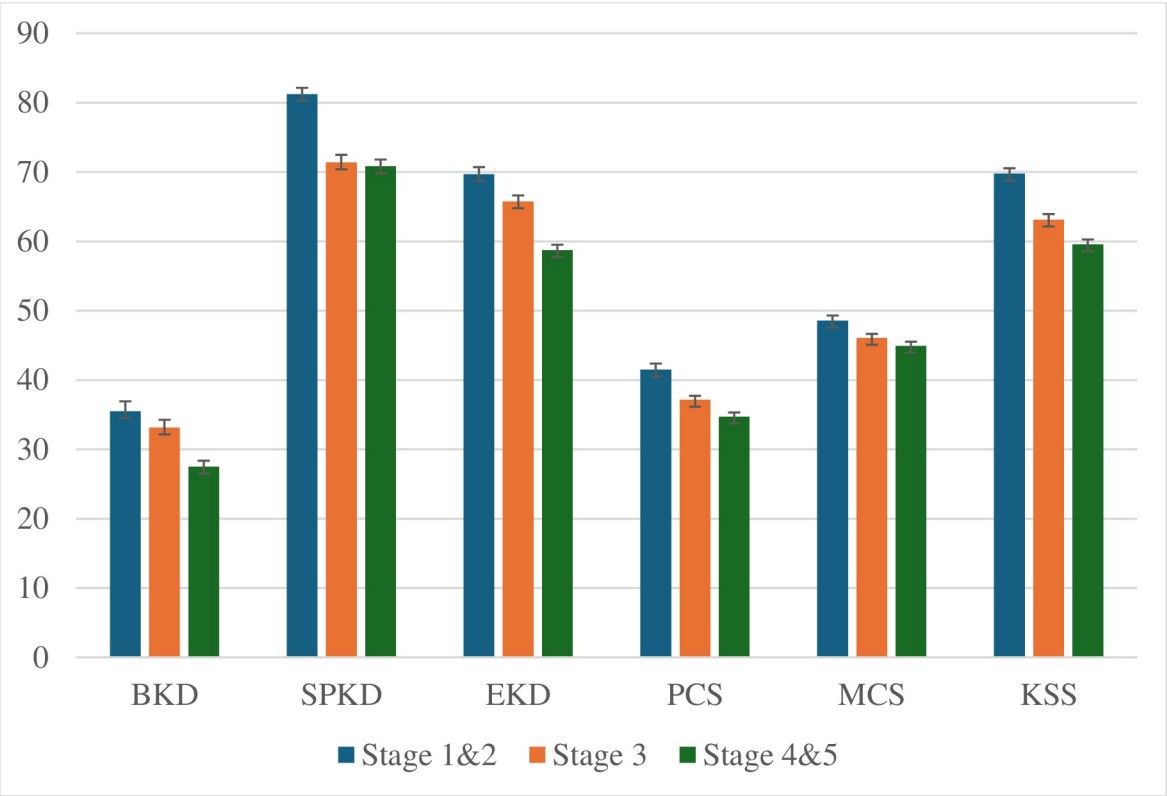

**Fig 1. Quality of life of the CKD patients according to KDQoL-36 domine at different stages of CKD.** BKD: burden of kidney disease, SPKD: symptom/problem of kidney disease, EKD: effects of kidney disease, PCS: physical component summary, MCS: mental component summary, and KSS: KDQoL-36 Summary Score.

**Table 4. Multiple linear regression analysis for predictors of KDQoL-36, kidney disease specific quality of life domains.**

| Variable | Burden of Kidney Disease | | Symptom/Problem List | | Effects of Kidney Disease | | KDQoL-36 Summary Score (KSS) | |
|---|---|---|---|---|---|---|---|---|
| | Coefficients (95% CI) | p-value | Coefficients (95% CI) | p-value | Coefficients (95% CI) | p-value | Coefficients (95% CI) | p-value |
| **Gender**: Female | | | | | -2.17[-4.88 – 0.53] | 0.116 | -1.59[-3.71 – 0.51] | 0.139 |
| **CKD stage** | | | | | | | | |
| Stage 1 & 2 | 1.0 | | 1.0 | | 1.0 | | 1.0 | |
| Stage 3 | -1.17[-4.45 – 2.1] | 0.482 | -5.82[-8.48 – (-3.16)] | <0.001 | -2.61[-5.12 – (-0.11)] | 0.040 | -3.99[-6.07 – (-1.92)] | <0.001 |
| Stage 4 &5 | -5.33[-8.71 – (-1.94)] | 0.002 | -6.32[-9.05 – (-3.59)] | <0.001 | -8.46[-11.03 – (-5.89)] | <0.001 | -6.83[-8.98 – (-4.69)] | <0.001 |
| **Age (Years)** | | | | | | | | |
| 18–44 | 1.0 | | 1.0 | | 1.0 | | 1.0 | |
| 45–64 | -2.33[-5.27 – 0.61] | 0.120 | -0.33[-2.74 – 2.07] | 0.784 | -2.47[-4.74 – (-0.20)] | 0.033 | -1.65[-3.47 – 0.15] | 0.073 |
| ≥65 | -7.57[-11.6 – (-3.55)] | <0.001 | -2.13[-5.44 – 1.17] | 0.206 | -8.03[-11.15 – (-4.91)] | <0.001 | -5.03[-7.54 – (-2.52)] | <0.001 |
| **Marital Status**: Married | - | | - | | -3.73[-6.04 – (-1.42)] | 0.002 | - | |
| **Occupation** | | | | | | | | |
| Job Service | 1.0 | | 1.0 | | 1.0 | | 1.0 | |
| Housewife | -4.04[-7.90 – (-0.18)] | 0.040 | -6.62[-10.11 – (-3.13)] | <0.001 | -0.78[-4.34 – 2.80] | 0.670 | -3.26[-6.11 – (-0.41)] | 0.025 |
| Farmer/Day Laborer | -0.90[-5.06 – 3.24] | 0.668 | -3.20[-6.68 – 0.28] | 0.072 | -0.45[-3.74 – 2.83] | 0.786 | -1.61[-4.27 – 1.04] | 0.233 |
| Retired | 0.05[-6.20 – 6.30] | 0.988 | -8.93[-14.0 – (-3.86)] | 0.001 | 1.45[-3.31 – 6.22] | 0.549 | -3.78[-7.74 – 0.18] | 0.062 |
| Others (Student, unemployed) | -4.72[-8.53 – (-0.90)] | 0.015 | -6.80[-9.99 – (-3.62)] | <0.001 | -0.49[-3.55 – 2.57] | 0.753 | -3.95[-6.36 – (-1.53)] | 0.001 |
| **Educational status** | | | | | | | | |
| Illiterate | 1.0 | | 1.0 | | 1.0 | | 1.0 | |
| Primary | 0.29[-3.62 – 4.21] | 0.883 | 3.28[0.05 – 6.50] | 0.046 | 0.95[-2.08 – 3.99] | 0.539 | 1.70[-0.80 – 4.20] | 0.184 |
| Secondary | 2.18[-2.11 – 6.48] | 0.318 | 3.31[-0.33 – 6.95] | 0.075 | 1.26[-2.16 – 4.69] | 0.468 | 1.91[-0.86 – 4.69] | 0.176 |
| Higher Education | 0.56[-4.05 – 5.18] | 0.810 | 3.52[-0.36 – 7.41] | 0.075 | 0.39[-3.28 – 4.06] | 0.834 | 1.54[-1.43 – 4.52] | 0.309 |
| **Higher income** (≥ 20000 Tk) | - | | -2.42[-4.92 – 0.06] | 0.056 | 1.91[-0.57 – 4.39] | 0.131 | - | |
| Long duration of CKD (≥24 months) | -6.36[-8.91 – (-3.81)] | <0.001 | 4.22[2.10 – 6.33] | <0.001 | -3.07[-5.06 – (-1.08)] | 0.003 | - | |
| **Co-morbidity** | | | | | | | | |
| Absence of diabetics | 1.89[-0.71 – 4.50] | 0.154 | -0.05[-2.42 – 2.31] | 0.964 | 1.05[-1.16 – 3.28] | 0.351 | 0.68[-1.17 – 2.54] | 0.467 |
| Absence of hypertension | 0.77[-2.14 – 3.70] | 0.601 | 0.48[-2.04 – 3.02] | 0.705 | -1.40[-3.79 – 0.98] | 0.247 | 0.20[-1.77 – 2.18] | 0.839 |
| Absence of CVD | - | | 3.27[0.35 – 6.20] | 0.028 | 0.85[-1.89 – 3.59] | 0.542 | 1.82[-0.45 – 4.11] | 0.117 |
| ≥3 Medication used | - | | -11.37[-14.01 – (-8.72)] | <0.001 | -4.93[-7.42 – (-2.44)] | <0.001 | -7.19[-9.23 – (-5.15)] | <0.001 |
| Constant | 39.47[33.18 – 45.76] | <0.001 | 81.32[75.06 – 87.59] | <0.001 | 75.57[69.28 – 81.85] | <0.001 | 71.58[66.75 – 76.42] | <0.001 |

CKD: chronic kidney disease, CVD: cardiovascular disease, CI = confidence interval

Abeywickrama *et al* [39] reported a much lower SPKD compared to other components of HRQoL. The symptom burden often depends upon patients' perspectives on their physical and psychological suffering. Also, the difference may be due to various rates of complication, CKD stages, a different assessment tool used (SF-36, KDQoL-36) different socioeconomic statuses and the number of medications used. The PCS and MCS were found 37.19 ± 8.1, 45.94 ± 7.5 respectively, in this study. Those values are similar to previous findings [40, 41]. As observed in other studies, the mental domains of QoL were less impacted than the physical domains, indicating that renal patients often experience lower physical QoL compared to

**Table 5. Multiple linear regression analysis for predictors of KDQoL-36, physical and mental domains.**

| Variable | SF-12 Physical (PCS) | | SF-12 mental (MCS) | |
|---|---|---|---|---|
| | Coefficients (95% CI) | p-value | Coefficients (95% CI) | p-value |
| **CKD stage** | | | | |
| Stage 1 & 2 | 1.0 | | 1.0 | |
| Stage 3 | -3.05[-4.93 – (-1.16)] | 0.002 | -1.05[-2.87 – 0.76] | 0.255 |
| Stage 4 &5 | -5.64[-7.58 – (-3.70)] | <0.001 | -2.54[-4.41 – (-0.66)] | 0.008 |
| **Age (Years)** | | | | |
| 18–44 | 1.0 | | 1.0 | |
| 45–64 | 0.94[-0.71 – 2.59] | 0.266 | -0.46[-2.05 – 1.12] | 0.567 |
| ≥65 | -2.49[-4.79 – (-0.19)] | 0.034 | -2.86[-5.02 – (-0.71)] | 0.009 |
| **Marital Status** | | | | |
| Others | - | | 1.0 | |
| Married | | | 2.13[0.53 – 3.72] | 0.009 |
| **Occupation** | | | | |
| Job Service | 1.0 | | - | |
| Housewife | -1.16[-3.56 – 1.24] | 0.343 | | |
| Farmer/Day Laborer | -1.01[-3.37 – 1.34] | 0.399 | | |
| Retired | -4.44[-8.01 – (-0.87)] | 0.015 | | |
| Others (Student, unemployed) | 0.40[-1.82 – 2.63] | 0.722 | | |
| **Educational status** | | | | |
| Illiterate | - | | 1.0 | |
| Primary | | | 2.08[-0.09 – 4.26] | 0.061 |
| Secondary | | | 2.53[0.17 – 4.89] | 0.035 |
| Higher Education | | | 3.49[1.05 – 5.93] | 0.005 |
| **Income** | | | | |
| Lower income (<20000 TK) | 1.0 | | - | |
| Higher income (≥ 20000 Tk) | -0.79[-2.49 – 0.89] | 0.356 | | |
| **Duration of CKD** | | | | |
| Low (<24 months) | 1.0 | | - | |
| Long (≥24 months | 1.90[0.40 – 3.39] | 0.013 | | |
| **Diabetics** | | | | |
| Present | 1.0 | | 1.0 | |
| Absent | 0.50[-1.15 – 2.16] | 0.550 | 1.68[0.05 – 3.30] | 0.043 |
| **Hypertension** | | | | |
| Present | 1.0 | | 1.0 | |
| Absent | 0.05[-1.74 – 1.85] | 0.950 | 1.63[-0.10 – 3.36] | 0.065 |
| **CVD** | | | | |
| Present | 1.0 | | 1.0 | |
| Absent | 1.29[-0.77 – 3.36] | 0.219 | 2.35[0.36 – 4.34] | 0.020 |
| **Medication used** | | | | |
| <3 Medication | 1.0 | | 1.0 | |
| ≥3 Medication | -3.70[-5.55 – (-1.85)] | <0.001 | -0.09[-1.87 – 1.69] | 0.921 |
| Constant | 40.73[36.95 – 44.50] | <0.001 | 40.76[36.89 – 44.63] | <0.001 |

CKD: chronic kidney disease, CVD: cardiovascular disease, CI = confidence interval

mental QoL [11, 15, 40, 41]. This could probably be due to the chronic nature of the disease with patients psychologically adapting to challenges over time, which is reflected in their mental health assessments. Physical symptom management, such as pain relief and mobility support, can help improve PCS scores, while psychosocial interventions like counseling and stress management can enhance MCS scores. A multidisciplinary care approach, combining both physical and mental health strategies, could significantly improve overall HRQoL in CKD patients.

The stages of CKD (severity of disease) have a significant impact on quality of life and have been documented as an important predictor of HRQoL of all domains of KDQoL-36 as well as kidney-targeted health-related quality of life (KSS). As the patients approached renal failure, their HRQoL measurements declined, in contrast to the early stages of the illness. Aggarwal *et al* [42] and Kefale *et al* [40] supported these findings by finding that people with advanced renal disease had the lowest HRQoL scores. Sharma *et al* [43] demonstrated that the decline in kidney function (as determined by eGFR) was correlated with the decline in HRQoL in all areas, with the most severe impairment observed in CKD stage 5. The data presented by Pei *et al* [44] suggests that HRQoL experiences a substantial decline in CKD patients, which can be employed to predict their future mortality. Thus, in earlier stages, focusing on preventive measures, such as lifestyle interventions and medication optimization, could slow progression and preserve HRQoL. In contrast, for patients in advanced stages, more intensive interventions both medical and psychosocial could be prioritized to manage symptoms and improve patients' overall well-being. Recognizing the declining HRQoL as a marker of disease progression could facilitate timely, personalized interventions, ensuring that patients receive the appropriate care at each stage.

In this study, CKD patients who were younger had better QoL than those who were older in terms of all components of KDQoL. This finding was consistent with a study conducted in Australia, which reported that younger CKD patients had significantly better QoL than older ones [45]. This finding is consistent with a few studies performed in the State of Palestine [46]; Oman [47], India [42], Nepal [48] etc., which reported that older age was associated with poor HRQoL. Older people with advanced CKD are more likely to experience frailty, multi-morbidity, and polypharmacy that contribute to impaired quality of life and reduced survival compared with younger adults with the same diagnosis [49]. Therefore, older CKD patients require personalized treatment plans that focus on managing frailty, multi-morbidity, and polypharmacy. Early identification of these risks and a holistic approach addressing both physical and psychosocial health can improve clinical outcomes and QoL. Recognizing age-related factors allows healthcare providers to better meet the diverse needs of CKD patients.

Unemployment resulting from disease, challenges in participating in one's existing job, and failure to fulfill societal expectations might contribute to a decrease in HRQoL. According to Senanayake *et al* [50], having a greater level of education, being employed, and having a higher income were all important factors that independently predicted better scores on the summary component. A few other studies have also shown positive association of higher education, employment status and economic condition with HRQoL components [36, 40, 48, 51]. Greater education levels often translate into better economic conditions and ultimately more access to information and better health-seeking behavior. The results of the present study indicate that occupation, rather than income, is a direct predictor of KDQoL. CKD patients who are retired, housewives, or students likely do not contribute to the family income and are less likely to have the financial means to pay for adequate healthcare, leading to a lower quality of life (HRQoL). On the other hand, working in the private sector, possibly indicates socioeconomic and/or professional incentives resulting in better health opportunities [51]. To improve HRQoL in CKD patients with limited resources, strategies should prioritize access to

subsidized healthcare, financial support, and community resources, while integrating health education and psychological care. A holistic, patient-centered approach combining medical and social support is key to enhance their overall well-being.

When compared, patients with CVD and diabetes, mental component score and symptoms/problems were significantly lower than those with no history of CVD and diabetes. This is in agreement with Aggarwal et al [42], who found PCS, MCS, and scores of all subscales to be significantly lower for patients with CVD and diabetes. Senanayake et al [38] reported that the presence of comorbidities was a significant predictor of having high symptom burden scores. CKD patients suffer from high levels of inflammation, which is a shared risk factor for both CVD and diabetes, potentially modulating HRQoL related to CKD [52]. Early detection and management of CKD, along with addressing comorbid conditions like diabetes mellitus and cardiovascular disease, can help mitigate the impact on patients' health-related quality of life. Based on these findings, treatment strategies should focus on integrated care, prioritizing the management of both CKD and its comorbidities. A multidisciplinary approach involving nephrology, cardiology, and endocrinology can help optimize glycemic control, manage hypertension, and reduce inflammation. For patients with prolonged disease duration, addressing both physical symptoms and mental health is crucial, with tailored interventions that include psychosocial support to alleviate the high symptom burden and improve overall well-being.

Over medication in kidney failure patients was linked to depression [53]. Adjeroh et al [54] found that polypharmacy lowers HRQoL in non-dialysis CKD patients. Berhe et al [36] had mentioned that pill burden indicated for the management of different CKD-related complications may also cause an additional burden to patients' HRQoL. The present study also found that >3 medication prescriptions predicted lower PCS and practically all KDQoL subscale scores. As kidney function declines, CKD requires many pharmacological prescriptions to control disease-related consequences like anemia, metabolic problems, hyperlipidemia, and mineral and bone abnormalities [55]. CKD impairs glomerular filtration, tubular secretion, and reabsorption. Most drugs are eliminated through the kidneys, so reduced kidney function changes drug pharmacokinetic (absorption, distribution, metabolism, and excretion) and pharmacodynamic (drug-receptor interactions) properties, increasing the risk of life-threatening toxicities [56]. The greater the number of complications and advanced stage of CKD, the greater the number of medications, and hence the greater the burden on patients and hence the invariably poor HRQoL [57]. Therefore, it is crucial for healthcare providers to carefully monitor medication regimens in CKD patients to minimize the risk of adverse effects and optimize therapeutic outcomes. Additionally, patient education on medication management and adherence is essential to improve HRQoL in this population.

The implications of the current study should be considered in relation to its design strengths and limitations. The study's exhaustive examination of numerous patient health characteristics may be an advantageous feature, as it has the potential to predict the HRQoL of the study population. The KDQoL-36 is a validated and accurate instrument for assessing HRQoL. The present study investigated the HRQoL in all stages of CKD who are non RRT, an area of focus previously ignored. The study's shortcomings, such as its cross-sectional design, prevented the assessment of temporal impacts. Conducting longitudinal research is essential for obtaining a more comprehensive understanding of the factors associated with HRQoL in CKD patients. Sampling bias may limit the generalizability of the study's findings. The exclusion of RRT patients ensures a focus on non-RRT CKD patients, which is the study's primary objective. However, the findings are not applicable to individuals at more advanced stages of CKD or those undergoing RRT. Future studies should aim to include a more representative sample that spans all stages of CKD, including those on RRT, to improve the generalizability

of the findings and provide a more comprehensive view of HRQoL across the full spectrum of CKD stages. Finally, while the study identifies several potential predictors of HRQoL, future research should also address additional confounding factors such as medication burden, drug interactions, and medication adherence. By addressing these limitations, future studies will be better positioned to offer more robust and actionable insights, which can ultimately inform more effective treatment strategies and interventions aimed at improving HRQoL in CKD patients.

## Conclusion

This study highlights critical determinants of health-related quality of life (HRQoL) in chronic kidney disease (CKD) patients in Bangladesh, including advanced disease stage, older age, and comorbidities, with physical health being more severely impacted than mental health. These findings underscore the need for targeted, individualized interventions addressing physical symptoms and disease progression, especially for high-risk groups. Integrating routine HRQoL assessments into clinical practice can facilitate timely, cost-effective care by enabling early identification of at-risk patients and personalizing treatment strategies. Future efforts should prioritize a bench-to-bedside approach, translating these insights into scalable interventions such as nutritional counseling, psychosocial support, routine HRQoL monitoring, and optimized pharmacological management, tailored to resource-limited settings. Additionally, longitudinal studies should evaluate the progression of HRQoL across CKD stages and assess the impact of socioeconomic and cultural factors, ensuring interventions are context-specific and sustainable. By bridging the gap between research and clinical application, these strategies hold the potential to enhance patient-centered care locally while serving as a model for other low-resource environments globally.

## Supporting information

**S1 Appendix. Kidney Disease Quality of Life (KDQoL) data calculation and data underlying the results presented in the study.**
(XLS)

## Acknowledgments

We would like to thank Kidney Foundation Hospital and Research Institute (KFHRI) as they helped us to obtain some recent data. Also, thanks to the data collection team. Special thanks to University of Dhaka for the research grant scheme for Dhaka University Teacher (2023–24) that is financially supported by the University Grant Commission (UGC), Bangladesh.

## Author Contributions

**Conceptualization:** Tasmia Tasnim, Kazi Muhammad Rezaul Karim, Tanjina Rahman, Harun-Ur Rashid.

**Data curation:** Tasmia Tasnim, Kazi Muhammad Rezaul Karim, Tanjina Rahman, Harun-Ur Rashid.

**Formal analysis:** Kazi Muhammad Rezaul Karim, Tanjina Rahman.

**Funding acquisition:** Kazi Muhammad Rezaul Karim.

**Methodology:** Tasmia Tasnim, Kazi Muhammad Rezaul Karim.

**Resources:** Tanjina Rahman.

**Software:** Kazi Muhammad Rezaul Karim.

**Supervision:** Tasmia Tasnim, Tanjina Rahman, Harun-Ur Rashid.

**Validation:** Kazi Muhammad Rezaul Karim.

**Visualization:** Kazi Muhammad Rezaul Karim, Harun-Ur Rashid.

**Writing – original draft:** Tasmia Tasnim, Kazi Muhammad Rezaul Karim.

**Writing – review & editing:** Tasmia Tasnim, Kazi Muhammad Rezaul Karim, Tanjina Rahman, Harun-Ur Rashid.

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
