## [Decision Letter · Decision Letter 0]

12 Nov 2024

PONE-D-24-41348Health-related quality of life and its predictors among chronic kidney disease patients: A hospital-based cross-sectional studyPLOS ONE

Dear Dr. karim,

Thank you for submitting your manuscript to PLOS ONE. After careful consideration, we feel that it has merit but does not fully meet PLOS ONE’s publication criteria as it currently stands. Therefore, we invite you to submit a revised version of the manuscript that addresses the points raised during the review process.

We look forward to receiving your revised manuscript.

Kind regards,

Ari Samaranayaka, PhD

Academic Editor

PLOS ONE

Journal Requirements:

“The project was completed under the research grant scheme for Dhaka University Teacher (2023-24) that is financially supported by the University Grant Commission (UGC), Bangladesh.”

Additional Editor Comments (if provided):

• Abstract says logistic regression was used in the analysis. But not presented any results from that analysis. Main text says multivariate regression was done, what analyses done here is unclear, please elaborate.

• Noted the conclusions were generalized to Bangladeshi CKD population. However the recruitment of participants was from a single hospital. How representative these participants to the Bangladeshi CKD population?

• Authors presented a formula used to calculate the sample size, need to define each notation in that formula. Appears the sample size was determined to estimate a proportion with a pre-specified precision. But there is no such an estimation in analysis or results sections. Instead, analysis appeared used linear regression, it is also different to the logistic regression mentioned in the abstract. Therefore sample size estimation incompatible with the analysis done.

• Noted the number of participants was same as the estimated sample size. How did you select that number of people from all eligible patients in that hospital in that period?

• Main source of the data was questionnaires (in addition extracting from medical records). What is the mode of interviews (online?, face to face?, assisted? If so by whom?, etc). Measures taken to mitigate possible bias (eg, social desirability bias) introduced by that?

• Main outcome measure is the KDQoL score along with its dimension scores. Could you explain these scores? Eg, what is/are the possible ranges of these scores? Higher scores indicate better or worse HRQoL?

• Statistical method section says SPSS was used for statistical analysis. Then later says Stata was used for multivariable regression. Does that mean multivariable analysis is not a statistical analysis? Throughout the manuscript authors used the terms multivariate and multivariable interchangeably. These two types of analyses are very different, they haven’t done a multivariate analysis.

• Page 14. Typo. “in sufficient” instead of “insufficient”.

• Tables 4 and 5. Minus sign and “to” are both denoted by “-“ when presenting confidence intervals. This is confusing.

• What is new in this study?

• Conclusion. “Younger age was associated with lower scores in all HRQoL subscales”. Where are evidence for that statement?

Reviewers' comments:

Reviewer's Responses to Questions

**Comments to the Author**

1. Is the manuscript technically sound, and do the data support the conclusions?

Reviewer #1: Yes

Reviewer #2: Yes

2. Has the statistical analysis been performed appropriately and rigorously? 

Reviewer #1: Yes

Reviewer #2: Yes

3. Have the authors made all data underlying the findings in their manuscript fully available?

Reviewer #1: Yes

Reviewer #2: Yes

4. Is the manuscript presented in an intelligible fashion and written in standard English?

Reviewer #1: Yes

Reviewer #2: Yes

5. Review Comments to the Author

Reviewer #1: Simple, easily understandable, appropriately narrated and explained, and conclusion matched with the objectives and purposes.

It would be informative and useful to narrate in introduction, discussion and conclusion how would this study outcome can be translated into focused tailoring of the treatment plan, bench to bedside. There should be some suggestions for future studies based on the outcome of this study.

Reviewer #2: The study examines the health-related quality of life (HRQoL) in chronic kidney disease (CKD) patients, using the Kidney Disease Quality of Life 36-item (KDQoL-36) questionnaire. The authors employed a cross-sectional design to assess HRQoL in CKD patients attending a hospital in Dhaka, Bangladesh. This design is appropriate for evaluating HRQoL at a single point in time, but as a cross-sectional study, it limits the ability to draw causal conclusions. The use of the KDQoL-36 questionnaire is well-justified, as it is a validated and widely used tool to measure HRQoL in kidney disease patients.The study provides valuable insights into the factors that predict HRQoL in CKD patients in Bangladesh, which could help inform patient care and interventions. However, methodological limitations—including the cross-sectional design, lack of detailed statistical reporting, and the potential for sampling bias—reduce the robustness of the conclusions.Recommend addressing limitations, confounders and clear explanations to further strengthen the article acceptance.

6. PLOS authors have the option to publish the peer review history of their article (what does this mean?). If published, this will include your full peer review and any attached files.

Reviewer #1: **Yes: **Emran Bin Yunus

Reviewer #2: **Yes: **Krishna Baradhi

---

## [Author Response · Author response to Decision Letter 0]

23 Dec 2024

Manuscript Number: PONE-D-24-41348

Health-related quality of life and its predictors among chronic kidney disease patients: A hospital-based cross-sectional study

PLOS ONE

Dear Editors and Reviewers, 

Thank you for your thorough review. Kindly take note that we have highlighted the changes in colors. Below are point to point responses to the Reviewers’ and Editor’s comments. 

Response to Reviewer 1

1. Is the manuscript technically sound, and do the data support the conclusions?- Yes

Author response: Thank you very much for your comments.

2. Has the statistical analysis been performed appropriately and rigorously?- Yes

Author response: Noted

3. Have the authors made all data underlying the findings in their manuscript fully available?- Yes

Author response: Noted

4. Is the manuscript presented in an intelligible fashion and written in standard English?- Yes

Author response: Noted

5. Simple, easily understandable, appropriately narrated and explained, and conclusion matched with the objectives and purposes. It would be informative and useful to narrate in introduction, discussion and conclusion how would this study outcome can be translated into focused tailoring of the treatment plan, bench to bedside. There should be some suggestions for future studies based on the outcome of this study.

Author response: Suggestions have been added in the last part of discussion and also in the conclusion part.

Response to Reviewer 2

1. Is the manuscript technically sound, and do the data support the conclusions?- Yes

Author response: Thank you very much for your comments.

2. Has the statistical analysis been performed appropriately and rigorously?- Yes

Author response: Noted

3. Have the authors made all data underlying the findings in their manuscript fully available?- Yes

Author response: Noted

4. Is the manuscript presented in an intelligible fashion and written in standard English?- Yes

Author response: Noted

5. The study examines the health-related quality of life (HRQoL) in chronic kidney disease (CKD) patients, using the Kidney Disease Quality of Life 36-item (KDQoL-36) questionnaire. The authors employed a cross-sectional design to assess HRQoL in CKD patients attending a hospital in Dhaka, Bangladesh. This design is appropriate for evaluating HRQoL at a single point in time, but as a cross-sectional study, it limits the ability to draw causal conclusions. The use of the KDQoL-36 questionnaire is well-justified, as it is a validated and widely used tool to measure HRQoL in kidney disease patients. The study provides valuable insights into the factors that predict HRQoL in CKD patients in Bangladesh, which could help inform patient care and interventions. However, methodological limitations—including the cross-sectional design, lack of detailed statistical reporting, and the potential for sampling bias—reduce the robustness of the conclusions. Recommend addressing limitations, confounders and clear explanations to further strengthen the article acceptance.

Author response: We acknowledge that this study has some limitations, including its cross-sectional design, which prevents causal inferences. Potential confounders like socioeconomic status, cultural factors, and psychological influences may have affected HRQoL scores but were not fully controlled for. Additionally, while the KDQoL-36 is widely used, its relevance in the Bangladeshi context may be limited, and further adaptation or qualitative methods could provide more comprehensive insights. Acknowledging these limitations, future research should focus on longitudinal studies and consider additional factors to better understand HRQoL in CKD patients.

Response to Editor’s comments:

Author response: Done Accordingly.

2. Thank you for stating in your Funding Statement:“The project was completed under the research grant scheme for Dhaka University Teacher (2023-24) that is financially supported by the University Grant Commission (UGC), Bangladesh.”Please provide an amended statement that declares *all* the funding or sources of support (whether external or internal to your organization) received during this study, as detailed online in our guide for authors at http://journals.plos.org/plosone/s/submit-now. Please also include the statement “There was no additional external funding received for this study.” in your updated Funding Statement. Please include your amended Funding Statement within your cover letter. We will change the online submission form on your behalf.

Author response: Done Accordingly.

Author response: Done Accordingly.

4. We note that your Data Availability Statement is currently as follows: [All relevant data are within the manuscript and its Supporting Information files.] Please confirm at this time whether or not your submission contains all raw data required to replicate the results of your study. Authors must share the “minimal data set” for their submission. PLOS defines the minimal data set to consist of the data required to replicate all study findings reported in the article, as well as related metadata and methods (https://journals.plos.org/plosone/s/data-availability#loc-minimal-data-set-definition).

Author response: Done Accordingly.

5. Abstract says logistic regression was used in the analysis. But not presented any results from that analysis. Main text says multivariate regression was done, what analyses done here is unclear, please elaborate.

Author response: It will be multiple linear regression and has been corrected in manuscript.

6. Noted the conclusions were generalized to Bangladeshi CKD population. However, the recruitment of participants was from a single hospital. How representative these participants to the Bangladeshi CKD population?

Author response: We recruited study participants from the Kidney Foundation Hospital and Research Institute, Bangladesh, a specialized renal care facility where several renowned nephrologists and consultants provide care. This institution serves patients from various districts across Bangladesh, reflecting the diverse demographics of the country's chronic kidney disease (CKD) population. The incidence of CKD in Bangladesh is estimated at 418,831.07 cases per year. The outpatient services at the hospital accommodate approximately 250 to 300 patients daily, all of whom are Bangladeshi residents. Given this, our study participants can be considered representative of the Bangladeshi CKD population.

This section has been added in the method section.

7. Authors presented a formula used to calculate the sample size, need to define each notation in that formula. Appears the sample size was determined to estimate a proportion with a pre-specified precision. But there is no such an estimation in analysis or results sections. Instead, analysis appeared used linear regression, it is also different to the logistic regression mentioned in the abstract. Therefore sample size estimation incompatible with the analysis done.

Author response: Sample size has been recalculated according to linear regression model.

8. Noted the number of participants was same as the estimated sample size. How did you select that number of people from all eligible patients in that hospital in that period?

Author response: This part has been revised in the method section.

9. Noted the number of participants was same as the estimated sample size. How did you select that number of people from all eligible patients in that hospital in that period?

Author response: This part has been revised in the method section.

10. Main source of the data was questionnaires (in addition extracting from medical records). What is the mode of interviews (online? face to face? assisted? If so by whom? Etc.). Measures taken to mitigate possible bias (eg, social desirability bias) introduced by that?

Author response: It was face to face interview. Trained interviewers collected the data. This part has been improved in the revised manuscript.

11. Main outcome measure is the KDQoL score along with its dimension scores. Could you explain these scores? Eg, what is/are the possible ranges of these scores? Higher scores indicate better or worse HRQoL?

Author response: Detailed information has been added in the revised manuscript. Higher scores indicate the better HRQoL. The Kidney Disease Quality of Life (KDQoL) instrument is a tailored adaptation of HRQoL tools designed for CKD patients. It combines generic HRQoL measures, such as physical and mental health, with CKD-specific domains, including symptoms, disease burden, effects on daily life, patient satisfaction, and work status. Higher KDQoL scores indicate fewer or less severe symptoms, though CKD patients generally score lower than the general population due to the disease's significant impact.

12. Statistical method section says SPSS was used for statistical analysis. Then later says Stata was used for multivariable regression. Does that mean multivariable analysis is not a statistical analysis? Throughout the manuscript authors used the terms multivariate and multivariable interchangeably. These two types of analyses are very different, they haven’t done a multivariate analysis

Author response: It has been corrected in the revised manuscript according to your suggestion.

13. Page 14. Typo. “in sufficient” instead of “insufficient”.

Author response: Done according to your suggestion.

14. Tables 4 and 5. Minus sign and “to” are both denoted by “-“ when presenting confidence intervals. This is confusing.

Author response: It has been corrected according to your suggestion.

15. What is new in this study?

Author response: Thank you for your thoughtful comments. This study makes a unique and innovative contribution by addressing the underexplored area of health-related quality of life (HRQoL) among chronic kidney disease (CKD) patients in Bangladesh, a low-income country with distinct socio-economic and healthcare challenges. Unlike most existing research, which predominantly focuses on high-income settings, this study provides valuable insights from a resource-limited context. By employing the KDQoL™-36 questionnaire and its summary score (KSS), we offer a comprehensive evaluation of physical, mental, and kidney-specific domains of HRQoL. To our knowledge, this is the first study in Bangladesh to apply these tools, enabling a multidimensional understanding of CKD’s impact.

The findings also highlight several novel predictors of HRQoL, such as the influence of employment status, education, marital status, and polypharmacy, alongside clinical factors like CKD stage and comorbidities. These insights provide a foundation for targeted, patient-centered interventions to improve HRQoL in CKD patients. Furthermore, our study addresses critical gaps in global CKD research by contributing context-specific evidence that may inform healthcare policies in similar resource-constrained settings. We believe these factors collectively underscore the significance of our work and its potential to improve CKD care and outcomes.

16. Conclusion. “Younger age was associated with lower scores in all HRQoL subscales”. Where are evidence for that statement?

Author response: The statement was corrected as younger age was associated with higher scores in all HRQoL subscales.

17. While revising your submission, please upload your figure files to the Preflight Analysis and Conversion Engine (PACE) digital diagnostic tool. To use PACE, you must first register as a user. Registration is free. Then, login and navigate to the UPLOAD tab, where you will find detailed instructions on how to use the tool. If you encounter any issues or have any questions when using PACE, please email PLOS at figures@plos.org. Please note that Supporting Information files do not need this step.

Author response: Done Accordingly.

---

## [Decision Letter · Decision Letter 1]

28 Jan 2025

Health-related quality of life and its predictors among chronic kidney disease patients: A hospital-based cross-sectional study

PONE-D-24-41348R1

Dear Dr. karim,

We’re pleased to inform you that your manuscript has been judged scientifically suitable for publication and will be formally accepted for publication once it meets all outstanding technical requirements.

Kind regards,

Ari Samaranayaka, PhD

Academic Editor

PLOS ONE

Additional Editor Comments (optional):

Reviewers' comments:

Reviewer's Responses to Questions

**Comments to the Author**

1. If the authors have adequately addressed your comments raised in a previous round of review and you feel that this manuscript is now acceptable for publication, you may indicate that here to bypass the “Comments to the Author” section, enter your conflict of interest statement in the “Confidential to Editor” section, and submit your "Accept" recommendation.

Reviewer #2: All comments have been addressed

2. Is the manuscript technically sound, and do the data support the conclusions?

Reviewer #2: Yes

3. Has the statistical analysis been performed appropriately and rigorously? 

Reviewer #2: I Don't Know

4. Have the authors made all data underlying the findings in their manuscript fully available?

Reviewer #2: Yes

5. Is the manuscript presented in an intelligible fashion and written in standard English?

Reviewer #2: Yes

6. Review Comments to the Author

Reviewer #2: Overall, this study makes a valuable contribution to understanding the factors that influence HRQoL among CKD patients, particularly in a Bangladesh-based context. The findings offer important insights into how various socio-demographic, medical, and psychological factors affect patients' quality of life. However, the cross-sectional design and the limited focus on certain psychological and social aspects suggest that further research is needed to deepen our understanding and improve patient care.

7. PLOS authors have the option to publish the peer review history of their article (what does this mean?). If published, this will include your full peer review and any attached files.

Reviewer #2: **Yes: **krishna baradhi

---

## [Editor Report · Acceptance letter]

30 Jan 2025

PONE-D-24-41348R1 

PLOS ONE

Dear Dr. karim, 

I'm pleased to inform you that your manuscript has been deemed suitable for publication in PLOS ONE. Congratulations! Your manuscript is now being handed over to our production team.

Kind regards, 

on behalf of

Dr. Ari Samaranayaka 

Academic Editor

PLOS ONE